# Mast Cells Differentiated in Synovial Fluid and Resident in Osteophytes Exalt the Inflammatory Pathology of Osteoarthritis

**DOI:** 10.3390/ijms23010541

**Published:** 2022-01-04

**Authors:** Priya Kulkarni, Abhay Harsulkar, Anne-Grete Märtson, Siim Suutre, Aare Märtson, Sulev Koks

**Affiliations:** 1Department of Pathophysiology, Institute of Biomedicine and Translational Medicine, University of Tartu, Ravila 19, 50411 Tartu, Estonia; priya.kulkarni@ut.ee (P.K.); abhay.harsulkar@ut.ee (A.H.); 2Department of Pharmaceutical Biotechnology, Poona College of Pharmacy, Bharati Vidyapeeth University, Erandwane, Pune 411038, India; 3Department of Pharmacology and Therapeutics, University of Liverpool, Liverpool L69 3BX, UK; a.martson@liverpool.ac.uk; 4Department of Anatomy, Institute of Biomedicine and Translational Medicine, University of Tartu, Ravila 19, 50411 Tartu, Estonia; siim.suutre@ut.ee; 5Department of Traumatology and Orthopaedics, Institute of Clinical Medicine, University of Tartu, L Puusepa 8, 51014 Tartu, Estonia; Aare.Martson@kliinikum.ee; 6Clinic of Traumatology and Orthopaedics, Tartu University Hospital, L Puusepa 8, 51014 Tartu, Estonia; 7Perron Institute for Neurological and Translational Science, Nedlands, WA 6009, Australia; 8Centre for Molecular Medicine and Innovative Therapeutics, Murdoch University, Murdoch, WA 6150, Australia

**Keywords:** immune cell differentiation, mast cells, osteophytes, osteoarthritis, proteomics, RNA-seq, synovial fluid

## Abstract

Introduction: Osteophytes are a prominent feature of osteoarthritis (OA) joints and one of the clinical hallmarks of the disease progression. Research on osteophytes is fragmentary and modes of its contribution to OA pathology are obscure. Aim: To elucidate the role of osteophytes in OA pathology from a perspective of molecular and cellular events. Methods: RNA-seq of fully grown osteophytes, collected from tibial plateau of six OA patients revealed patterns corresponding to active extracellular matrix re-modulation and prominent participation of mast cells. Presence of mast cells was further confirmed by immunohistochemistry, performed on the sections of the osteophytes using anti-tryptase alpha/beta-1 and anti-FC epsilon RI antibodies and the related key up-regulated genes were validated by qRT-PCR. To test the role of OA synovial fluid (SF) in mast cell maturation as proposed by the authors, hematopoietic stem cells (HSCs) and ThP1 cells were cultured in a media supplemented with 10% SF samples, obtained from various grades of OA patients and were monitored using specific cell surface markers by flow cytometry. Proteomics analysis of SF samples was performed to detect additional markers specific to mast cells and inflammation that drive the cell differentiation and maturation. Results: Transcriptomics of osteophytes revealed a significant upregulation of mast cells specific genes such as chymase 1 (CMA1; 5-fold) carboxypeptidase A3 (CPA3; 4-fold), MS4A2/FCERI (FCERI; 4.2-fold) and interleukin 1 receptor-like 1 (IL1RL1; 2.5-fold) indicating their prominent involvement. (In IHC, anti-tryptase alpha/beta-1 and anti- FC epsilon RI-stained active mast cells were seen populated in cartilage, subchondral bone, and trabecular bone.) Based on these outcomes and previous learnings, the authors claim a possibility of mast cells invasion into osteophytes is mediated by SF and present in vitro cell differentiation assay results, wherein ThP1 and HSCs showed differentiation into HLA-DR+/CD206+ and FCERI+ phenotype, respectively, after exposing them to medium containing 10% SF for 9 days. Proteomics analysis of these SF samples showed an accumulation of mast cell-specific inflammatory proteins. Conclusions: RNA-seq analysis followed by IHC study on osteophyte samples showed a population of mast cells resident in them and may further accentuate inflammatory pathology of OA. Besides subchondral bone, the authors propose an alternative passage of mast cells invasion in osteophytes, wherein OA SF was found to be necessary and sufficient for maturation of mast cell precursor into effector cells.

## 1. Introduction

Osteoarthritis (OA) is a degenerative disorder characterised by progressive erosion of articular cartilage along with the other pro-inflammatory and degenerative conditions. The disease is a major contributor to worldwide disability in the elderly population. Owing to the complex and elusive nature, the treatment options in OA are limited to palliative pain management and surgically fitted implants under terminal conditions. A general understanding of the disease pathology can be presented as a vicious circle of oxidative stress promoting inflammation and inflammation accentuating oxidative stress leads to a pathological degeneration of joint tissues including articular cartilage, meniscus and subchondral bone [1]. Chronic low-grade synovial inflammation is now accepted as one of the fundamental causes of OA [1], wherein synovial cells and articular chondrocytes are primary sources of cytokines such as interleukin-1 beta, tumour necrosis factor-α. Besides these cells, infra-patellar fat pad, which is situated in the space between the patellar tendon, femoral condyle, and tibial plateau and covered with synovial membrane, serve as an additional source of the disease-specific cytokines [2]. However, cellular and molecular mechanism underlining this inflammation has not been elucidated completely. Hyper-regulation of immunity in the form of macrophages and a range of pro-inflammatory factors secreted by the cells has been attributed as the driving factors of OA [3].

Osteophytes, commonly known as ‘bone spurs’, are a hallmark of OA joints. These are marginal ectopic formations of osteo-cartilaginous metaplastic tissue mostly at the junction of periosteum and synovium that appear to merge with or overgrown with the original articular cartilage [4]. Although, osteophytes do not necessarily warrant any clinical intervention, depending on the position they can cause nerve compression in the spine and more friction in the knee joints that lead to crepitus, discomfort and pain and may require surgical resection [4]. Clinically, osteophytes define the structural progression of OA along with the other clinical signs including joint space narrowing, subchondral sclerosis and cartilage defects.

Blom and colleagues have extensive research work on osteophytes [4,5,6] and have described the process of osteophytes formation (known as osteophytosis) elaborately. During osteophytosis, mesenchymal stem cells at the joint differentiate into chondrocytes showcasing different signatures of chondrogenesis. Further, centrally placed well differentiated hypertrophic chondrocytes subsequently undergo osteoblastogenesis and facilitate the growth of osteophytes. This entire process is similar to endo-chondral ossification that takes place in the foetal growth plate formation [4,5]. A fully grown and mature osteophyte is a bony outgrowth with a cartilage cap and minuscule bone marrow [4]. Macrophages, which are harbingers of synovial inflammation, play an essential role in the release of transforming growth factor-β (TGF-β), bone morphogenic protein-2 (BMP-2) and BMP-4, the key players involved in the osteophyte’s formation process. Experimental depletion of synovial macrophages results in a significant reduction in osteophytes, despite of a high dose of intra-articulate injections of TGF-β, as shown in the murine model [5]. In vivo studies showed that inhibition of TGF-β or over-expressing TGF-β antagonist prevents osteophyte formation [7,8]. In recent times, in fluorescent genetic cell-labelling and tracing mouse models contributions of fluorescently labelled cells to osteophytes formation after 2 or 8 weeks and their molecular identity were analysed using histology, immunofluorescence staining and RNA in situ hybridization. Osteophytes were found to be derived from platelet-derived growth factor receptor-A (PDGFRα)-expressing progenitor cells in periosteum and synovium that are descendants from growth differentiation factor-5 (GDF5)-expressing embryonic joint interzone. The authors further showed that SRY-box transcription factor-9 (SOX9) progenitor cells in periosteum contributed hybrid skeletal cells to the early osteophyte, while proteoglycan-4 (PRG4)-expressing progenitors from synovial lining supplied cartilage capping to the osteophyte, but not to bone [9].

Osteophytes have been investigated with high interest for their chondrogenic potential to decipher in vivo mechanisms of cartilage regeneration. However, analysis of structural compositions of newly formed matrix revealed differences from a normal make-up of high-quality hyaline articular cartilage [10]. In another view, osteophytosis is proposed as a repair mechanism in degenerating joints since the osteophytes at the right place may help to stabilize the joint. This is because animal studies have shown that surgical removal of the osteophytes can destabilize the joints [11]. However, the random appearance of the osteophytes contradicts this purview. Further, their association with symptoms such as restricted joint movement, discomfort and pain put them in a pathological perspective and questions their biomechanical benefits. Osteophytes, therefore, are a pathological feature in OA which are worthy of investigation in a molecular and cellular context.

Transcriptome analysis of a fully grown osteophyte is never attempted to gain a deeper insight into the pathophysiology associated with them. The authors performed a genome-wide transcriptome analysis of six paired samples, which included osteophytes and non-osteophytic bony tissue (as a control) obtained from the medial plateau of the tibia in OA patients. This investigation revealed patterns corresponding to extra-cellular matrix (ECM) re-modulation, tissue damage and recruitment and activation of immune cells such as mast cells in the osteophyte samples. The involvement of mast cells in OA pathology has been indicated by several studies [12,13,14]. The presence of mast cells and their degranulation products were reported in OA synovium and synovial fluid (SF) [12,14]. Based on the transcriptomics results, the authors further explored an association between osteophytes and mast cells, which is yet to be a studied topic. Furthermore, they provide evidence for mast cell presence in osteophytes of OA joints and demonstrate their maturation mediated by SF in temporal fashion with disease progression.

## 2. Results

### 2.1. Differential Transcripts between Osteophytes and Control Specimens

After statistical analysis, 595 genes were differentially expressed between osteophytes and non-osteophytic controls (logFC ≥ 2 and logFC ≤ −2). Out of 595 genes, 322 genes were found up-regulated (logFC ≥ 2), while 273 were down-regulated (logFC ≤ −2) (Figure 1a–c). To maintain brevity of the topic only up-regulated genes are described and discussed. In order to find out highly up-regulated genes, k-means clustering analysis was performed, wherein optimum number of clusters was determined by the elbow method. For up-regulated genes, optimal K was 7. Here, cluster numbers 1, 5 and 7 showed high significance (*p* < 0.001) and were considered for further analysis (Figure 2a,b). Among 322 up-regulated genes, 87 genes were found with a marked increase in their expression levels (*p* < 0.001). All 87 up-regulated genes and 22 down-regulated genes are listed with their *p*-value and fold change in Appendix A.

### 2.2. Functional Annotation of the Differentially Expressed Gene Networks

To find the functional relationship among the differentially expressed up-regulated genes, we performed a pathway enrichment analysis. Enrichment analysis is a popular method to identify biological themes in the complex lists of differentially expressed genes, using a context of prior knowledge. For this, we used Enrichr software, which contains 180, 184 annotated gene sets from 102 gene set libraries at present [15].

The enrichment analysis indicated a significant activation of matrix metalloproteinase as a top enriched canonical pathway with a combined score of 17.59 (Table 1). Additionally, osteoblast signalling (score 16.86), receptor activator of nuclear factor-kappa-Β ligand (RANKL/NFκB) signalling pathway (score 16.71), angiotensin-converting enzyme (ACE) inhibitor pathway (score 16.47) and osteoclast signalling pathway (score 15.73) were the other highly enriched pathways (Table 1). The inflammatory response pathway (WP453) provided an indication of immune-regulation and signalling. In brief, the functional analysis provided a footprint of bone and cartilage ECM remodelling and inflammatory responses, especially mediated by the canonical RANKL/NFκB signalling pathway.

### 2.3. Up-Regulated Genes

Up-regulated genes in the osteophytic tissue recorded several interesting observations from an OA pathological point of view. Prominently up-regulated genes were listed in Table 2. Significant up-regulation of CMA1 (5-fold) CPA3 (4-fold), membrane-spanning 4-domains A2/Fc fragment of IgE receptor 1a (MS4A2/FCERI; 4.2-fold) and interleukin 1 receptor-like 1(IL1RL1; 2.5-fold) demonstrated an active involvement of mast cells in osteophytes pathobiology. Additionally, osteophytes specimens were also characterized by a marked up-regulation of E selectin (SELE; 2.5-fold), collagen type-I alpha-1 chain (COL1A1; 2.04-fold) and COL1A2 (2.01-fold). Furthermore, MMP-1 (3.03-fold), MMP-3 (3.54-fold) and MMP-13 (3.2-fold), the key genes which mediate ECM remodelling, were also significantly up-regulated. To validate the transcriptomic findings, qRT-PCR was performed using CPA3, tryptase alpha/beta-1 (TPSAB1), CMA1, MMP-1, MMP-3 and MMP-13 genes and results are presented in Figure 3.

### 2.4. Immunohistochemistry (IHC) of the Osteophytes

Histopathological investigation of the osteophytic tissue revealed bone matrix with distinguishable osteoblasts, osteocytes and bone-matrix interspersed with occasional blood vessels arriving from the subchondral bone as well as the cartilage with distinguished columnar chondrocytes embedded in the cartilage matrix. IHC staining with mast cells specific antibodies anti-TPSAB1 and anti-FCERI revealed an extensive presence of mast cells in all the osteophyte samples Figure 4 (P1, P2, P3). Both antibody-stained cells were profoundly visible in the fibrocartilage and bony tissue of the osteophytes. The cells were grouped in large numbers along with the inner lining of the osteon as well as in the space between subchondral cancellous bone trabeculae in the osteophyte samples. Moreover, the cells were found localized along the chondro-osseous junction, where the osteophytic cartilage and bone meet.

Mast cell staining frequency was determined by the ordinal method as described [16]. Anti- TPSAB1 staining incidence range was 1–3% between control and the osteophyte samples except in one osteophyte sample (incidence range: 40%) (Figure 4 C, P1, P2, P3- anti-TPSAB1 panel). In this osteophyte sample, mast cells were extensively gathered in the space between subchondral cancellous bone trabeculae (Figure 4 anti-TPSAB1 panel). On the other hand, anti-FCERI staining incidence in control tissue ranged between 5–30%, it varied 10–70% for the osteophyte samples. After statistical analysis, both the anti-bodies staining showed a significant difference between control and osteophyte samples, *p* < 0.01 (Figure 4 histograms-control and osteophytes). IHC study observations corroborated with the expression signatures of the mast cells revealed in the transcriptome study.

### 2.5. In Vitro Cell Differentiation Assays

The 10th-day flow cytometry analysis of SF treated human monocytes (ThP1) cells showed a clear and significant differentiation into human leukocyte antigen- DR isotype (HLA-DR+) and CD206+ cells as compared to untreated control (UC) and phorbol-12-myristate-13-acetate (PMA) (positive control) (Figure 5a,b). Furthermore, a staining pattern for both the antibodies was found similar and the highest number of differentiated cells was observed after treating the cells with SF of Kellgren-Lawrence (KL) grade III (Figure 5a,b).

On the other hand, haematopoietic stem cells (HSCs) were stained with FCERI antibody after 9 days of SF treatment. Flow-cytometry analysis of HSCs revealed a significant differentiation into FCERI+ cells after SF treatment of KL grade III in comparison to UC, PMA (Figure 5c). Interestingly, the staining pattern of FCERI antibody in HSCs was matching to HLA-DR and CD206 staining in ThP1 cells (Figure 5a1,b1,c1). Outcomes of the in vitro cell differentiation assay, therefore, showed that OA SF is able to drive a clear differentiation of monocytes into macrophages and HSCs into mast cells.

### 2.6. Proteomics Analysis of OA SF

Since SF was observed to provide an immunomodulatory environment for the differentiation of immune cell precursors, a proteomic investigation of OA SF was carried out. For this, OA SFs (*n* = 16) from different grades were subjected to LC/MS/MS analysis after depletion for high abundance proteins. The expressed proteins were identified against UniProt (access date—13 November 2020; UniProt Consortium, Cambridge, UK) *Homo sapiens* reference proteome database. A grade-wise comparison revealed 799 differentially up-regulated proteins as compared to KL grade I SFs. Light chain immunoglobulins, pro-inflammatory S100 proteins, histones, actins, mitogen-activated protein kinase (MAPK) family and mast cells de-granulation proteases such as carboxypeptidase and cathepsins were the major subset of proteins found differentially up-regulated (Appendix A). A grade-wise picture of these proteins and key Reactome pathway functional analysis of these protein subsets is presented in Figure 6. The involvement of pathways such as the innate immune system, Fc-gamma receptor (FCGR)-dependent phagocytosis coupled with marked accumulation of light chain immunoglobulins were potentially suggested the mast cell activation. On the other hand, differentially up-regulated proteases such as carboxypeptidase, carboxypeptidase Q and cathepsins (L1, D, B and G) were signs of mast cell function.

## 3. Discussion

Transcriptome analysis of the osteophyte samples showed a footprint of bone and cartilage remodelling with significant up-regulation of MMP and osteoblast signalling pathways (Table 1) that involve cathepsins, COL1A1, COL1A2, SELE and MMPs transcripts. Cathepsin G (6.38-fold) and cathepsin K (2.24-fold) were among the significantly up-regulated cathepsins. Cathepsin K is best known for its active involvement in bone resorption and was found in osteoclasts [17]; it was detected in the zones of active bone remodelling in osteophytes along with the other cathepsins [18]. On the other hand, Cathepsin G is one of the proteases in pre-packaged granules of mast cells, which activates MMPs for ECM protein degradation [19]. It is also involved in the activation of osteoclasts and osteolytic lesions by mediating the release of a soluble form of RANKL from the bound RANKL present on the surface of osteoblasts [20]; the RANKL signalling pathway was one of the activated pathways in the functional analysis of expressed gene network in the osteophyte samples (Table 1). COL1A1 and COL1A2 are bone matrix proteins and their expression level defines the osteoblast maturation stage [21]. A significant up-regulation of both the genes in osteophyte samples indicated the presence of fully mature osteoblasts. Our understanding of osteophytes in OA is greatly influenced by Blom et al., 2004 and Gelse et al., 2012 [5,10]. As reported by both the authors, growth factors, especially TGFβ and BMPs are attributed to triggering and maintaining the growth of osteophytes; however, none of these growth factors were found up-regulated in our transcriptomics. Perhaps this was because the osteophytes collected for this study were fully grown and likely to be in the senescence phase as they were obtained from the patients undergoing knee replacement surgery and hence, were unlikely to reflect any developmental changes.

MMP-1 (3.02-fold), MMP-3 (3.53-fold) and MMP-13 (3.19-fold) were found prominently up-regulated in the osteophytes and were also validated by qRT-PCR (Figure 3). Of note, the control tissue quantity was limited and could not be included for qRT-PCR validation; this fact was recorded as a limitation of the present data. The role of MMPs in osteophytes is linked with multiple functions including ECM remodelling. Bone and ECM remodelling in osteophytes shows a mechanistic similarity with the process of endochondral ossification in growth plate formation; for example, proliferating chondrocytes express collagen type-II, whereas the hypertrophic chondrocytes express collagen type-X and MMP-13. On the other hand, regarding endochondral ossification, osteoblasts express MMP-13 and collagen type-I, whereas osteoclasts express MMP-9. MMPs are directly involved in the degradation of ECM components, collagenase or proteoglycan [22]. Furthermore, the expression of MMP-1 and MMP-3 was reported in different areas of osteophytic tissue [23]. Osteoblasts, isolated from osteophytes of OA joints were shown to produce MMP-13 and further predicted to contribute to the worsening of OA pathology [24]. Gelse et al., 2012 reported a presence of MMP-13 and MMP-9 in osteophytic cartilage [10]. Thus, our transcriptomics results for MMPs are in line with previous research.

CMA1 (5-fold), CPA3 (4.02-fold), MS4A2/FCERI (4.22-fold) and IL1RL1 (2.5-fold) were among the prominently up-regulated genes and were expressed in all the osteophyte samples of the study patients. Patient to patient variation of these genes is presented in Table 2. These genes are signatures of mast cells and ultimately indicated an active involvement of these cells in the molecular events associated with osteophytes. CMA1 is the only gene found in humans, which encodes a chymotryptic serine proteinase. CMA1 and CPA3 are common components of mast cell granules and were found involved in the degradation of endogenous proteins [25]. MS4A2/FCERI is a member of the membrane-spanning 4A gene family; the gene encodes a beta subunit, which is a high-affinity IgE receptor and provides a trigger for mast cell degranulation [26]. IL1RL1, a receptor for IL-33 cytokine, belongs to the toll-like receptor superfamily. IL-33 is synthesized de novo by activated mast cells and is recognized as a mediator of sterile inflammation. The cytokine is found augmented in many allergic diseases as well as in rheumatoid arthritis, psoriasis and atherosclerosis. More importantly, IL-33 binding to IL1RL1/ST2 induces differentiation, survival and chemotaxis of mast cells and further activates them to produce various cytokines [27,28].

In OA, mast cells are one of the major infiltrates of inflamed synovium along with macrophages and T cells [29]. Active participation of mast cells in OA pathology was confirmed by animal studies, wherein two distinct mast cell-deficient mice models [C57BL/6J-KitW-sh/W-sh (KitW-sh/W-sh) mice and Cpa3-Cre; Mcl-1fl/fl (Hello Kitty) mice] were found significantly protected against OA-related pathologies such as cartilage-loss, synovitis and osteophytes formation [13]. Furthermore, mast cell degranulation products such as CPA3 and CAM1 were detected in OA SF in substantial amounts and their concentrations were correlated with MMP-2 and MMP-9 levels [30]. The presence of CPA3 was documented in osteophytic cartilage [10]. This collective evidence successfully establishes the involvement of mast cells in OA. Our data further mark a significant existence of mast cells in osteophyte samples and present them as effector cells, contributing to the ECM and the bone remodelling process of osteophytes. Prominent mast cell activity in these samples can be interpreted in the form of significant up-regulation of phospholipase A2 group IIA (PL2G2A) (4.64-fold) and MMP-13 (3.19-fold), as the inflammatory repercussions. PLA2G2A is often called ‘inflammatory secretory phospholipase A2′ (sPLA2) and is involved in many inflammatory pathologies such as rheumatoid arthritis and psoriasis [31]. It encodes secreted phospholipase A2, which hydrolyse the Sn2 position of phospholipid molecules and usually release unsaturated fatty acids such as arachidonic acid that in turn, promotes biosynthesis of inflammatory prostaglandins [31]. Bingham III et al., 1999 reported the presence of PLA2G2A in mast cells granules [32], while Enomoto et al., 2000 demonstrated the redistribution of PLA2G2A from granules of resting mast cells to fusion sites and plasma membrane to facilitate exocytosis in mast cells of bone marrow origin [33]. Although PLA2G2A is secreted by other cell types such as platelets, neutrophils, macrophages, importantly, however, exogenous PLA2G2A is able to stimulate mast cell degranulation as shown by Murakami et al., 1993 in bone marrow-derived mast cells [34]. Enzymatic activity of MMP-13 in periapical lesions of inflammatory periodontitis was found correlated with the number of tryptase-positive mast cells, which were undergoing active remodelling similar to the osteophytes [35], while MMP-13 is a known indicator of inflammation in OA pathology [36]. In addition to the proteolytic activity of MMP-13, mast cell-specific tryptase and chymase are also responsible for bone and cartilage remodelling and release cartilage and cellular degradation products, which are similar to damage-associated molecular patterns (DAMPs). These cartilage and cellular breakdown products serve as endogenous agonists that interact with toll-like receptors and generate a sterile inflammatory response [37]. Consequently, osteophytes serve as a source of breakdown products that further promote the worsening of OA. The authors previously demonstrated a temporal relationship in the accumulation of glycosaminoglycan in SF with increasing severity of OA [38].

The transcriptome outcomes of this study indicated a link between osteophytes and mast cells; hence, the authors further attempted to explore the origin of mast cells in osteophytes. Given that, mast cell precursors (MCPs) are known to originate in bone marrow by haematopoiesis [39] and osteophytes do have minuscule bone marrow depending on its size and maturity, the subchondral bone is a logical route for mast cells invading into osteophytes. However, there is no published evidence endorsing this channel. Also, this route does not explain how MCPs get matured into effector mast cells. Nevertheless, our IHC study observation wherein, the antibodies-stained cells were seen in noticeable number in the space between subchondral cancellous bone trabeculae in the osteophyte samples, supports the purview that MCPs from subchondral bone marrow could be a source of mast cells in osteophytes. Additionally, IHC study of the osteophyte samples showed an extensive accumulation of anti-FCERI and anti-TPSAB1-stained cells in the cartilaginous region as compared to the subchondral bone, suggesting that the mast cells invade into osteophytes through this region and reach the tide mark of the cartilage-subchondral bone junction. Anti-TPSAB1-staining was also observed in zones of cartilage and subchondral bone, indicating extracellular tryptase in the tissue as a sign of degranulation of mast cells. The cartilaginous region of osteophytes is directly exposed to SF. Based on these IHC study outcomes; the authors want to claim a possibility that invasion of the mast cells into osteophytes is mediated by SF (Figure 7). They present the outcomes of in vitro cell differentiation assays performed on ThP1 and HSCs to support their claim.

Circulatory mast cell progenitors (MCPs) are the major source of mast cells in osteophytes. OA SF, which is composed of cytokines and mast cell regulatory factors such as histones, S100A12 and IgG kappa and lambda chain, play an important role in transformation of MCPs and monocytes into effector cells such as mast cells and macrophages, respectively. Responding to the chemotactic signals (CXCL 9, 10 and 11) mast cells migrate and populate osteophytes. Mast cell de-granulation factors such as CPA3, tryptase, chymase, cathepsins, and SELE are involved in elevated ECM remodelling of osteophytes. These degradation products and cell debris (DAMPs) accumulate in SF, which act as inducers for mast cells via the TLR-2/4 pathway. Increasing action of active mast cells and macrophages contribute to synovitis, tissue damage, and cartilage loss. This figure is created with BioRender.com (accessed date—18 August 2021). 

The hypothesis of the in vitro cell differentiation assay was to test the role of OA SF in the maturation of MCPs by providing the necessary cocktail of cytokines, cell degradation products, and immunoglobulins. In the previous studies, the authors showed that this cocktail can be used to induce inflammation in SW982, U937, HIG82 and ThP1 cell lines [1,40,41]. As demonstrated by these assays, the differentiation of HSCs and ThP1 cells into FCERI+ and HLA-DR+/CD206+ phenotypes, respectively, after SF treatment, indicate a decisive role of OA SF in the differentiation of polarised cells into effector cells such as mast cells and macrophages (Figure 5a–c). A comparatively lower number of differentiated FCERI+ cells after SF treatment should be interpreted on the basis of mast cells proportion in OA SFs as reported by other authors [42,43,44]. The percentage of mast cells in OA SF samples was reported as 0.3% of the total cell population [42]. SF samples from minimal OA and established OA revealed the cell percentage as 1.05 ± 0.97 % and 0.84 ± 0.82 % of the leukocytes in all fluids, respectively [43]. A similar number of mast cells was reported in OA SF samples by Malone et al., 1986 [44]. Thus, the proportion of differentiated FCERI+ cells after SF treatment as revealed in the in vitro assay was in line with other similar studies. Lastly, the maximum differentiation of ThP1 and HSCs into the effector cells was seen after the SF treatment of KL grade III. This is likely to be the effect of the highest inflammatory milieu present in the KL grade III SF samples as reported by the authors in their previous publications [40]. On the other hand, this inflammatory milieu was found reduced in the KL grade IV SF samples as an effect of near-total worn-out cartilage in advanced OA [41] and is a possible reason for the relatively lower number of differentiated cells after the SF treatment of KL grade IV samples.

Proteomics analysis of the SF samples, which was used to treat the ThP1 cells and HSCs, revealed differentially expressed mast cell regulatory proteins such as Ig light chains, free histones and S100 proteins (Appendix A). Ig light chain proteins (kappa and lambda) have the capacity to bind to FCERI and FCgRI receptors on mast cells that ultimately trigger a hypersensitive response [45]. Similarly, S100A12 at low levels act as a chemotactic agent for mast cells, while their high concentration triggers degranulation of the cells (Figure 6) [46,47]. Mitogen-activated protein kinase (MAPK) proteins, free histones, and high mobility group box-1 (HMGB-1) are released in extracellular fluids as a result of the cytotoxic effect of augmented inflammation and extensive cell death and degeneration. Particularly, free histones are potent activators of immune cells by inducing inflammation via toll-like receptor-4 (TLR4) and TLR2 receptors [48,49] (Figure 6). Therefore, it is likely that dying cells release free histones in SF of OA joints, which ultimately elevate inflammation by stimulating various immune cells including mast cells. Interestingly, our thinking is supported by Tasaka et al., 1990, wherein exposure of rat peritoneal mast cells to a mixture of histones resulted in instant degranulation and the release of histamine, indicating that mast cells readily respond to free histones [50]. Furthermore, a significant elevation in MAS-related G protein-coupled receptor-X2 (MRGPRX2) (4.5-fold, in the transcriptome analysis) along with its receptor neurotensin (37.18-fold, as detected differentially expressed in SF proteomics analysis) is a strong indicator of the existing IgE independent pathway of mast cell activation in osteophyte samples. MRGPRX2 binds to cationic ligands, neuropeptides and opioids and, therefore, represents IgE independent pathway of activation. Both the pathways trigger distinct patterns of secretion of mast cell mediators [51]. The outcomes from IHC and in vitro cell differentiation assays provide a clear indication that besides subchondral bone, an alternate route for mast cells invasion into osteophytes exists and deeper confirmation studies along this line should take place (Figure 6). Furthermore, the significant up-regulation of SELE and chemokines such as chemokine ligand-9 (CXCL9), CXCL10 and CXCL11 was likely to be associated with increased mast cell deployment into osteophytes because of mast cell specificity of these chemokines [52].

Mast cells in osteophytes appear to be a new dimension of OA pathophysiology. Their invasion into cartilaginous region in a significant amount demands a deeper investigation from a pathological point of view. This is because, unlike other tissues, aneural and avascular cartilage is unlikely to produce any allergic or inflammatory response and therefore raises a question about the role of allergic or inflammatory mast cells in osteophytes.

## 4. Methods

### 4.1. Ethical Statement

All methods performed in this study abide by the Declaration of Helsinki and the study protocols used in this research were approved by the Institutional Ethics Committee of the University of Tartu (282/T13, 227/T-14 and 76785). All the recruited patients signed an informed written consent form to participate in the study.

### 4.2. Osteophyte Samples Collection

Osteophytes were collected from six knee OA patients during knee replacement surgery (Institutional Ethics approval number—282/T13). Demographic details of the patients from whom the osteophyte samples were collected are given in Appendix A, while the patient selection criteria for knee replacement surgery is described in the Appendix A). They were obtained from the medial condyle of the tibia. The authors decided to focus particularly on the tibial osteophytes because they are considered as one of the features of OA [53] and further indicate the disease severity [54]. Non-osteophytic tissue collected from epiphyseal trabecular bone from the lateral condyle of the tibia was used as a control.

OA diagnosis was based on clinical and radiological evaluation and was performed by an experienced orthopaedic surgeon. The disease severity (or OA grading) was determined using KL score. KL score system is based on radiological signs of OA; as per this system, grade I is doubtful narrowing of the joint space and possible, indistinguishable osteophyte. Grade II is definite clearly identifiable osteophytes and possible narrowing of the joint space. Grade III is with moderate multiple osteophytes with definite joint space narrowing, and grade IV is marked with large osteophytes with marked narrowing of joint space [38].

### 4.3. SF Collection

Our research group had collected approximately 100 SF samples from progressive grades of OA patients undergoing knee arthrocentesis procedure as described [38]. The fluid collection protocol was approved by the Institutional Ethics committee (BVDU/MC/01, 227/T-14 and 76785). The fluid collection process was performed in a minor operation theatre under strict aseptic conditions to collect SFs from early-grade OA patients. The fluids from advanced grade OA were collected at the time of knee replacement surgery. Patient selection criteria for SF collection is described in Appendix A as SD2. SF sample grading was the same as OA grading. A subset of SF samples from the repository available with the research group was used for in vitro cell differentiation assay and proteomics study as described later in this section. Demographic details of the patients, who were selected for SF collection are given in Appendix A. The SF samples from patients 1–12, were used for cell treatment during in vitro cell differentiation assay, while all SF samples from patients 1–16 were subjected to proteomics study.

### 4.4. RNA Extraction

Approximately, 50 mg of the collected sample (osteophytes and control) was homogenised using liquid nitrogen and 500 μL of Trizol reagent (Thermo Fisher Scientific Inc., Carlsbad, CA, USA) in mortar and pestle. The homogenate was centrifuged at 12,000× *g*, 10 min, at 4 °C and the supernatant was transferred into a new 2.0 mL tube and incubated for 5 min at RT. chloroform (100 μL) was then added to the sample, incubated at RT for 2 min and centrifuged at 12,000× *g* for 15 min at 4 °C; the aqueous phase was transferred into a new 2.0 mL tube. An equal amount of freshly prepared 70% ethanol was added to the sample and transferred into the RNeasy Mini spin column for further RNA isolation as per the manufacturer’s instructions (Qiagen, Valencia, CA, USA). Total RNA quality was evaluated using an Agilent 2100 Bioanalyzer with RNA 6000 Nano Kit (Agilent Technologies Inc., Santa Clara, CA, USA) and the quantity was estimated using a Qubit 2.0 fluorometer with the RNA HS Assay Kit (Thermo Fisher Scientific Inc., Carlsbad, CA, USA). The average RNA integrity number (RIN) of samples was between 2.8 and 7.7. The samples were stored at −80 °C until further processing.

### 4.5. Whole Transcriptome Analysis of the Osteophytes

Fifty nanograms of the total RNA was amplified by applying the Ovation RNA-Seq System V2 (NuGen, Emeryville, CA, USA) after which the resulting cDNAs were used to prepare the DNA fragment library with SOLiD System chemistry and barcode adaptors (Thermo Fisher Scientific Inc., Carlsbad, CA, USA). Prior to sequencing, all 12 libraries were labelled with different barcodes and were pooled together in equal amounts. Sequencing was performed using the SOLiD 5500W platform and fragment sequencing chemistry (Thermo Fisher Scientific Inc., Carlsbad, CA, USA).

Raw reads (75 bp) were colour-space mapped to the human genome hg19 reference using the Maxmapper algorithm implemented in the Lifescope software (Thermo Fisher Scientific Inc., Carlsbad, CA, USA). Mapping to multiple locations was permitted. The quality threshold was set to 10, giving the mapping confidence more than 90. Reads with a score of less than 10 were filtered out. The average mapping quality was 30. Analysis of the RNA content and gene-based annotation was conducted with the whole transcriptome workflow. 

After quality control of the samples, to perform differential gene expression analysis, non-normalized raw counts were used for the EdgeR package. EdgeR is a flexible tool for RNAseq data analysis to find differentially expressed genes. It performs model-based scale normalization, estimates dispersions and applies a negative binomial model. Further, it implements negative binomial model fitting followed by testing procedures for determining differential expression [55].

### 4.6. qRT-PCR Analysis for the Transcriptome Validation

To confirm the transcriptome analysis, qRT-PCR was performed using Applied Biosystems Step One Real-Time PCR System (Applied Biosystems, Foster City, CA, USA). For this, RNAs from the transcriptomics osteophyte samples were used. However, due to the unavailability of control tissue RNA, the qRT-PCR validation was limited to confirm the expression of key upregulated genes from the transcriptomics data in osteophytes. PureLink RNA Mini Kit (Ambion Inc., Invitrogen Co., Carlsbad, CA, USA) was used to isolate RNA from the cells and cDNA synthesis was performed using Superscript First-Strand Synthesis System (Invitrogen Co., Carlsbad, CA, USA) as per the manufacturer’s instructions. TaqMan gene expression assays used (Applied Biosystems, Foster City, CA, USA) to quantify the expression levels were CPA3 (gene expression assay ID—Hs00157019_m1), CMA1 (gene expression assay ID—Hs00156558_m1), TPSAB1 (gene expression ID—Hs02576518_gH), MMP1 (gene expression assay ID—Hs00899658_m1), MMP3 (gene expression assay ID—Hs00968305_m1) and MMP13 (gene expression assay ID -Hs0023392_m1). The amount of expressed genes was normalized to the amount of β-actin (ACTB) (gene expression assay ID—Hs01060665_m1). Cycling conditions were 50 °C for 2 min; 95 °C for 10 min; and 40 cycles of 95 °C for 15 s and 60 °C for 1 min. The data were analysed using Data Assist software (version 3.0).

### 4.7. IHC Analysis of the Osteophyte Samples

The osteophyte specimens were decalcified with Sakura TDE 30 Decalcifier System and were embedded in paraffin after fixation in formalin. Then, 5 µm sections were cut, deparaffinized and were treated with 0.9% H_2_O_2_ to inactivate endogenous peroxidase. The sections were then treated with Dako REAL Antibody Diluent (S2022; Dako Denmark A/S, Glostrup, Denmark) to block non-specific binding. After blocking, the sections were incubated with a mouse monoclonal antibody to TPSAB1 (MA5-11711, Thermofisher) and rabbit polyclonal antibody to FC epsilon RI (ab229889, Abcam, Waltham, MA, USA) to stain mast cells overnight at 4 °C. Primary antibody dilution was 1:200. Visualization of the primary antibody was performed using the commercial kit “Dako REAL EnVision DetectionSystem, Peroxidase/DAB+, Rabbit/Mouse” (K5007; Dako Denmark A/S, Glostrup, Denmark). Washing steps in between were performed using phosphate-buffered saline (PBS), containing 0.07% of Tween 20 as the detergent. Toluidine blue (Applichem, Darmstadt, Germany) was used for background staining. No immune staining was noted in negative controls, where the primary antibody was omitted. IHC images were obtained with Zeiss LSM-510 Confocal Laser -scanning microscope. Further, an ordinal method was used to calculate mast cells staining frequency as described [16]. An ordinal method is a semiquantitative method that reflects cellular staining frequency or intensity. Using this cellular frequency, we estimated the mast cells staining incidence (%) in the control tissues and in the osteophyte samples.

### 4.8. In Vitro Cell Differentiation Assay

To investigate the potential of OA SFs in inducing immune cell differentiation, in vitro cell differentiation assays were planned. In the assays, ThP1 and HSCs were treated with different grades of OA SFs for 9 days. The status of newly differentiated cells was analysed by relevant flow-cytometry markers. Here, ThP1 and HSCs were considered immune cell precursors.

#### 4.8.1. Cell Lines

##### ThP1 and HSCs

The ThP1 cell line was purchased from National Centre for Cell Science (NCCS), Pune, India, while freshly isolated, certified HSCs from human bone marrow were procured from Nirav Biosolutions, Pune, India for this assay. Both cell types were maintained in Roswell Park Memorial Institute (RPMI) 1640 medium (HiMedia Laboratories, LLC, Lincoln University, West Chester, PA, USA) + 10% foetal bovine serum (FBS) (HiMedia Laboratories, LLC, Lincoln University, West Chester, PA, USA) + 2 mmol/L L-glutamine + 100 U/mL penicillin + 100 µg/mL streptomycin (Sigma-Aldrich, St. Louis, MO, USA), at 95% relative humidity and 5% CO_2_ at 37 °C.

Both cell lines were seeded at a density of 1 × 10^5^ cells/mL and were treated with 10 % (of culture medium) SF of different OA grades. We used 3 SF samples of each KL grade; thus, altogether (4 KL grades × 3 SF samples), 12 SF samples were used for these assays. Cell differentiation was monitored for 9 days, during which a complete media change was performed every 3 days. The cells treated with PMA (Sigma-Aldrich, St. Louis, MO, USA) (dose—100ng/mL) were used as a positive control, while untreated cells were used as a negative control. On the 10th day, the adherent cells were collected for flow cytometry analysis. To harvest the adherent cells without enzymatic digestion, the cells were incubated in 0.5 mM ethylenediamine tetraacetic acid (EDTA) in Dulbecco’s phosphate-buffered saline for 15 min at 37 °C and 5% CO_2_. After 15 min, the cells were collected by repeated vigorous pipetting against the bottom of the 24-well plate.

The harvested cells were fixed using (−20 °C) methanol/acetone (1:1) (incubation time—10 min on ice) and were further blocked with human FC blocking solution (Human TrueStain FcX, BioLegend, San Diego, CA, US) as per the manufacturer’s instruction in order to prevent any non-specific antibody binding. ThP1 cells were stained with HLA-DR (PE-Cy5.5, Miltenyi Biotec, Bergisch Gladbach, Germany) and CD206, (PE, Miltenyi Biotec, Bergisch Gladbach, Germany) as per the manufacturer’s instructions. On the other side, HSCs were stained with FCERI (FITC, BioLegend, San Diego, CA, USA) as per the manufacturer’s instructions. For each marker, the median fluorescence intensity was measured by Attune Nxt Acoustic flow cytometer. All the flow cytometry experiments were performed in triplicate and repeated three times.

### 4.9. Proteomics Analysis of OA SF

The proteomics study was carried out using sixteen SFs; we included four samples of each grade (4 × 4 = 16).

For the depletion of high abundance albumin and immunoglobulins from SF samples, a protein depletion column was used as per the manufacturer’s instructions (proeteoMinerprotein Enrichment Small-Capacity Kit, BioRad, Hercules, CA, USA; catalogue No. 1633006). Later, protein precipitation was attained by mixing the sample with a mixture of 100% trichoroacetic acid, 0.4% deoxycholate (TCA + DOC) in a 1:3 (*v*/*v*) ratio for 20 min at 4 °C and later centrifuged for 15 min at 17,000 rpm. The supernatant was discarded and in the undisturbed pellet, three volumes (of the original sample volume) of RT 100% acetone were added, vortexed, incubated for 10 min at RT and centrifuged at 17,000 rpm for 15 min. The precipitates were air-dried on ice for 10 min until no residual liquid was visible. Each precipitated pellet was then suspended in 100 μL of 7 M urea, 2 M thiourea, 100 mM ammonium bicarbonate (ABC) solution (7/2 urea:thiourea buffer). After reduction and alkylation of cysteine bonds with 5 mM dithiothreitol and 20 mM chloroacetamide, respectively, for 1 h at RT in the dark, the samples were digested for 4 h in 1:50 (enzyme: Protein) ratio using *Lysobacter enzymogenes* (Wako Pure Chemical Industries, Richmond, VA, USA). Solutions were diluted five times with 100 mM ABC and further digested overnight at RT with 1:50 dimethylated porcine trypsin (Sigma-Aldrich). Digested samples were then desalted using reversed-phase C18 StageTips. Samples were reconstituted in 0.5% trifluoro acetic acid (TFA) for the subsequent LC/MS/MS analysis performed as described [56].

MS raw data were processed with the MaxQuant 1.4.0.8 software package [57]. Methionine oxidation, asparagine/glutamine deamidation and protein N-terminal acetylation were defined as variable modifications, while cysteine carbamidomethylation was set as a fixed modification. A peptide search was performed against in silico trypsin digested (C-terminal cleavage after lysine/arginine without proline restriction) UniProt (www.uniprot.org; accessed date—13 November 2020) *Homo sapiens* reference proteome database. First and main search MS mass tolerances were ±20 and ±4.5 ppm, respectively. MS/MS mass accuracy tolerance was ±20 ppm. Protein identifications were reported if ≥1 razor or unique peptides of ≥7 amino acids were identified. Transfer of peptide identifications (match between runs) based on accurate MS1 mass and RT was allowed. Protein quantification was reported if ≥1 peptide was quantified with ≥3 points. Label-free protein intensities were normalized using the MaxLFQ algorithm [58]. Peptide-spectrum match and protein false discovery rate (FDR) were kept ≤1% using a target-decoy approach. All other parameters were set as default.

## 5. Statistical Analysis

To find out the differential transcriptome between osteophytes and non-osteophytic control tissue, the authors used group-wise comparisons, where negative binomial fitting was followed by an exact test. False discovery rate (FDR) adjustment was used for multiple testing corrections. An FDR threshold of 0.05 was applied for statistical significance. In the case of qRT-PCR validation of the transcriptome, IHC study and in vitro cell differentiation assays, statistical analysis was carried out using GraphPad Prism 5 Software (San Diego, CA, USA) using one-way ANOVA followed by Bonferroni’s test for multiple comparisons. *p* values < 0.05 were considered significant. In the SF proteomics analysis, differentially expressed proteins were detected by performing a paired *t*-test on the grade-wise expressed proteins, where the proteins expressed in KL grade I SFs were used as a control group.

## Figures and Tables

**Figure 1 ijms-23-00541-f001:**
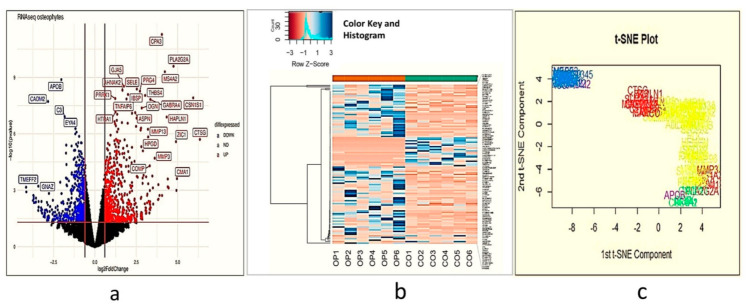
Demonstrate an overview of differentially expressed genes in transcriptome analysis performed using osteophyte samples and non-osteophytic control tissue obtained from six knee OA patients (*n* = 595). (**a**) Volcano plot of the differentially expressed genes. Here, black dots represent insignificant genes, while blue and red dots represent down-regulated and up-regulated genes, respectively. The plot is generated using LogFC values of the expressed genes. Carboxypeptidase A3 (CPA3), selectin E (SELE), membrane-spanning 4-domains A2/Fc fragment of IgE receptor 1a (MS4A2/FCERI), chymase 1 (CMA1), interleukin 1 receptor-like 1 (IL1RL1), collagen type 1 alpha 1 chain (COL1A1), COL1A2, matrix metalloproteinase 1 (MMP-1), MMP-3 and MMP-13 are among the significantly upregulated genes; (**b**) Histogram of the highly significant genes, which were obtained after K-means cluster analysis; (**c**) t-SNE plot of ‘highly differentiated’ genes (generated after K-means cluster analysis). t-SNE is a dimensional reduction technique of presenting large gene data sets; we generated a two-dimensional t-SNE plot to verify the cluster analysis. For this, we considered logFC, logCPR and *p*-values of the highly significant genes in the osteophyte specimens.

**Figure 2 ijms-23-00541-f002:**
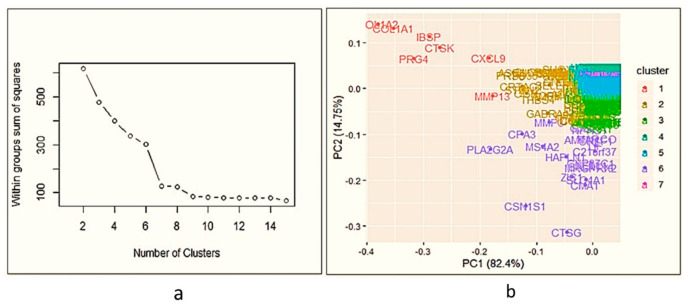
Cluster analysis of differential gene expression between osteophyte and control samples. (**a**) Denotes elbow method of k-means clustering to find out highly significant up-regulated; the optimum number of clusters is 7. (**b**) Cluster analysis of up-regulated genes—as k-means clustering for up-regulated is 7, the data is divided into 7 clusters. For up-regulated genes, clusters 1, 5 and 7 showed *p* < 0.001; 87 genes are highly up-regulated in osteophytes.

**Figure 3 ijms-23-00541-f003:**
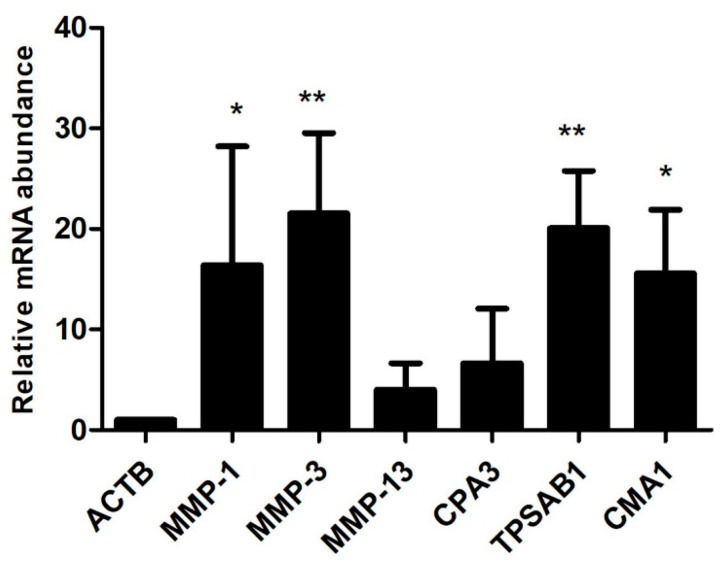
qRT-PCR analysis to validate transcriptome results of the osteophyte samples. mRNA levels of CPA3, CMA1, TPSAB1, MMP-1, MMP-3 and MMP-13 were normalized against ACTB, the housekeeping gene. All the values are expressed as mean ± SD. TPSAB1 (20.04-fold), CMA1 (15.57-fold), MMP-1 (16.32-fold) and MMP-3 (21.5-fold) showed a significant up-regulation. CPA3 (6.64-fold) and MMP-13 (4.01-fold) also showed up-regulation, which remained insignificant on statistical scale, however. This qRT-PCR validation is limited to confirm the expression of key upregulated genes from the transcriptomics data in the osteophytes. * *p* < 0.05, ** *p* < 0.01 compared to ACTB.

**Figure 4 ijms-23-00541-f004:**
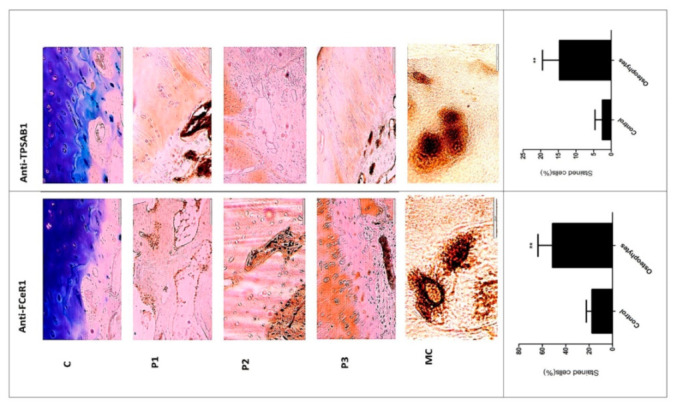
Representative slides of IHC staining with anti-FCERI and anti-TPSAB1 to visualize mast cells in osteophyte sections. Representative areas of osteophytes of study patients are shown as P1, P2 and P3. Antibody-stained mast cells were localized in bone trabeculae, cartilage region, where the columnar chondrocytes can be seen. MC are mast cells at 100X magnification revealing granular features. Bottom panel: histograms—control and osteophytes, denote anti-FCERI and anti-TPSAB1 stained cells (%) respectively; * *p* < 0.05, ** *p* < 0.01 compared to control.

**Figure 5 ijms-23-00541-f005:**
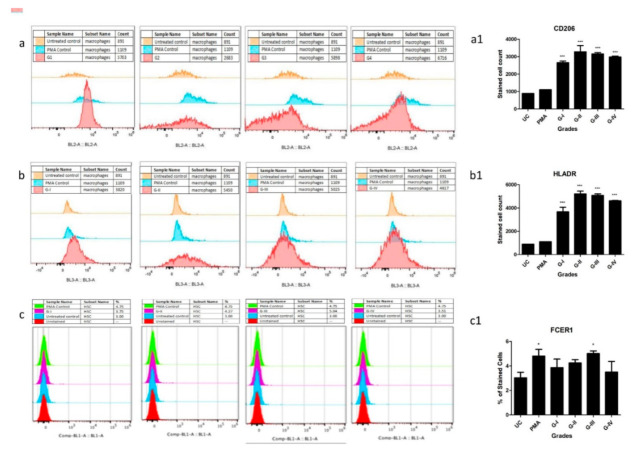
In vitro cell differentiation assay, where ThP1 and HSCs were treated with 10% (of culture medium) SF of KL grade I to IV for 9 days; expression of cell surface markers were analysed by flow cytometry on the 10th day. Shown are the representative overlaid histograms of the surface markers on ThP1 and HSCs. Bar graphs are the summary of the surface markers’ expression study, performed in triplicate. Sequence of ThP1 to macrophages differentiation analysed by (**a**) CD206, and (**b**) HLA-DR, (**a1**,**b1**) represent summary of CD206+ and HLADR+ expression values on ThP1 to macrophages differentiation sequence respectively; (**c**) sequence of HSCs to mast cells differentiation analysed by FCERI, (**c1**) represent summary of FCERI+ expression values on HSCs to mast cell differentiation sequence; * *p* < 0.05, *** *p* < 0.001 compared to UC.

**Figure 6 ijms-23-00541-f006:**
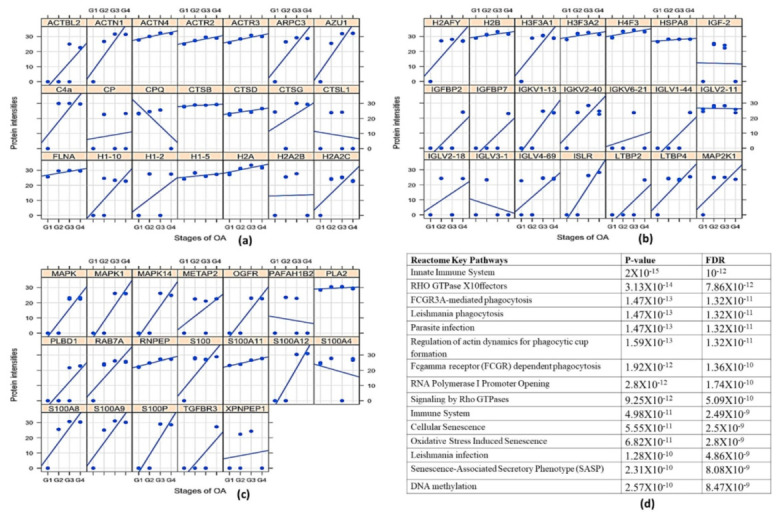
Proteomics analysis of OA SF. For the proteomics analysis, four patients of each grade (4 × 4 = 16) were analysed through MS/MS. (**a**–**c**) represents a grade-wise picture (lattice plots) of different protein subsets that were differentially expressed in OA SFs; differential protein expressions were determined by comparing the fold change of each protein found in KL grade I SFs. In total, 799 proteins were found differentially expressed. These proteins contain mast cell regulatory factors such as light chain immunoglobulins, S100 A12, histones, actins, mitogen-activated protein kinase (MAPK) family and mast cells degranulation proteases such as carboxypeptidase and cathepsins. (**d**)—functional analysis of the proteomics study in the form of key Reactome pathways with their *p*-values and FDRs.

**Figure 7 ijms-23-00541-f007:**
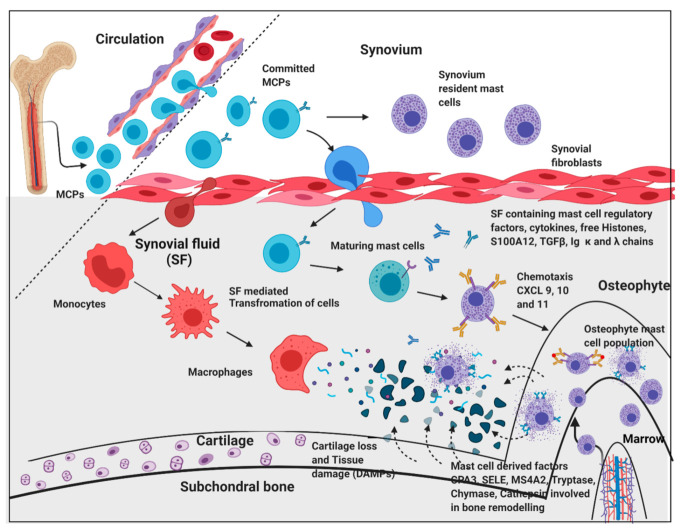
An overview of pathogenic dimension around osteophytes pertinent to mast cell action.

**Table 1 ijms-23-00541-t001:** Key pathways with their *p*-values, Z-score and combined score, generated during function analysis of the differentially expressed genes in osteophyte samples.

Sr. No	Name of the Pathway (with WikiPathway No.)	*p*-Value	Adjusted *p*-Value	Z-Score	Combined Score
1	Matrix metalloproteinases WP129	0.0005743	0.01594	−2.36	17.59
2	Osteoblast signaling WP322	0.00005487	0.004648	−1.72	16.86
3	RANKL/RANK (receptor activator of NFKB (ligand)) signaling pathway WP2018	0.0002299	0.008505	−1.99	16.71
4	ACE inhibitor pathway WP554	0.003794	0.07020	−2.95	16.47
5	Osteoclast signaling WP12	0.00008374	0.004648	−1.68	15.73
6	Photodynamic therapy-induced NF-kB survival signaling WP3617	0.0009077	0.02015	−1.81	12.69
7	Inflammatory response pathway WP453	0.01159	0.1040	−2.65	11.80
8	Oncostatin M signaling pathway WP2374	0.005383	0.07468	−1.93	10.06
9	Composition of lipid particles WP3601	0.04800	0.2424	−3.27	9.92
10	GABA receptor signaling WP4159	0.01235	0.1040	−2.18	9.60

**Table 2 ijms-23-00541-t002:** Prominently up-regulated genes in the osteophyte samples; patient to patient variation revealed by these mast cell-specific markers in the transcriptome analysis, wherein LogFC showed the difference of the gene expression between OA cases and controls and IfcSE was a standard deviation of the difference.

Gene Name	LogFC	IfcSE
CMA1	5.01	1.92
CPA3	4.02	0.59
MS4A2	4.22	0.63
IL1RL1	2.5	0.80

## Data Availability

Raw sequencing data with appropriate experimental information is available at the NCBI Gene-Expression Omnibus (GEO) repository under the accession number GSE66511.

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
