# Peer review of "Mast Cells Differentiated in Synovial Fluid and Resident in Osteophytes Exalt the Inflammatory Pathology of Osteoarthritis"

_ijms, 2022, doi:10.3390/ijms23010541_

Round 1

Reviewer 1 Report

Mast cells differentiated in synovial fluid and resident in osteophytes exalt the inflammatory pathology of osteoarthritis

Following are comments for improvement

  1. Avoid abbreviation in abstracts
  2. Need to add a completed detail on material and methodology in a separate section of materials before results and discussion.
  3. The reason for moving for cluster no. 1, 5 and 7 need to justify, hence need to explain why it is showing higher P
  4. Figures 1 and 2 need to be cited completely (not in part) in the text to discuss the clear significance of them.
  5. Similar to above, Table 1 has not been cited and discussed in the text.
  6. Figure 2(b) has very small fonts (no visibility).
  7. Authors have used many abbreviations in manuscript without explain their full form
  8.  The reference of “.. Blom et al., 2004 and Gelse et al., 2012 [3, 6]”is seems interesting to note, however authors need to add the way both authors define osteophyte formation process. Add recent review articles related to viscosupplements like 10.1016/j.cmpb.2020.105644
  9. Why few references are of red – coloured (41, 42, 43). Authors need to justify how these references correlate Figure 4C results.
  10.   Discussion part needs to be focused to draw  useful conclusions.

Author Response

Reviewers’ response to Reviewer 1:  

Comment 1: Avoid abbreviation in abstracts

Authors’ response: The authors thank the reviewer for his/her suggestion. All the abbreviations in the abstract are removed now.

Comment 2: Need to add a completed detail on material and methodology in a separate section of materials before results and discussion.

Authors’ response: The suggested change has been implemented now. We have moved ‘material and methodology’ section before results and discussion. More details on the methodology have been added.

Comment 3: The reason for moving for cluster no. 1, 5 and 7 need to justify, hence need to explain why it is showing higher P

Authors’ response:

All the up-regulated genes (logFC≥ 2) were subjected to K-means clustering analysis to find out highly up-regulated genes, wherein optimum number of clusters was determined by the elbow method. For up-regulated genes, optimal K was 7. The cluster number 1, 5 and 7 showed P< 0.001, which means the genes in these cluster had higher P values as compared to the other clusters. For the functional analysis, the authors focused on the highly up-regulated genes in the cluster 1, 5 and 7.  

Comment 4: Figures 1 and 2 need to be cited completely (not in part) in the text to discuss the clear significance of them.

Authors’ response: In light of the reviewer’s comment, Figure 1 and Figure 2 are revised in order to cite them in full and their significance is discussed in the results (line 305 -311).

Comment 5: Similar to above, Table 1 has not been cited and discussed in the text.

Authors’ response: The authors would point out humbly that Table 1 has been cited and discussed in the result section (line numbers- 321, 324, 388, 397).

Comment 6: Figure 2(b) has very small fonts (no visibility)

Authors’ response: The visibility of Figure 2 b has been improved by increasing font size.

Comment 7: Authors have used many abbreviations in manuscript without explain their full form

Authors’ response: We have added full form of all the abbreviations used in the manuscript text.

Comment 8: The reference of “.. Blom et al., 2004 and Gelse et al., 2012 [3, 6]”is seems interesting to note, however authors need to add the way both authors define osteophyte formation process. Add recent review articles related to viscosupplements like 10.1016/j.cmpb.2020.105644

Authors’ response: The authors thank the reviewer for his/her suggestion. The description of osteophytes formation process as defined by Blom and his colleagues is now added in the manuscript text (line numbers 100-111). However, we humbly want to point out that Gelse et al., 2012 has not defined the osteophytes formation process and referred the research work by Blom and his colleagues for the same.  

Regarding quoting the suggested publication on viscosupplements, the authors did not find any overlap with the of the present manuscript, the said reference is therefore not cited.  

Comment 9: Why few references are of red – coloured (41, 42, 43). Authors need to justify how these references correlate Figure 4C results.

Authors’ response: The references, which are left in red colour is a typo. The authors apologize for this. Please note that, in the revised version of the manuscript, Figure 4 is now Figure 5. Thus, the justification on correlating the results of Figure 5C and these references has been added now (line numbers 486-492).

Comment 10: Discussion part needs to be focused to draw useful conclusions.

Authors’ response: The discussion section has been revised and further improved in light of this and other reviewers.

Reviewer 2 Report

In this study, the authors attribute an important role in the inflammation and pathological processes of OA to Mast Cells present in osteophytes, indicating their source and mode of maturation. Very interesting, complete and well executed work. 

About in vitro cell differentiation assay, with regard to the maturation of MCs challenged with LS of different grades, shows a decrease going from G3 to G4, and being the difference of G3, the only statistically significant, perhaps it should be better discussed (the authors in the discussion rightly refer to the lower percentage of MCs vs Macrophages).

Author Response

Reviewers’ response to Reviewer 2: 

In this study, the authors attribute an important role in the inflammation and pathological processes of OA to Mast Cells present in osteophytes, indicating their source and mode of maturation. Very interesting, complete and well executed work. 

About in vitro cell differentiation assay, with regard to the maturation of MCs challenged with LS of different grades, shows a decrease going from G3 to G4, and being the difference of G3, the only statistically significant, perhaps it should be better discussed (the authors in the discussion rightly refer to the lower percentage of MCs vs Macrophages).

Authors’ response: The authors thank the reviewer for his/her appreciation on the research topic and the study design. In light of his/her comment, the decline in the differentiated FCERI+ cells after G4 SF treatment is explained in discussion of the manuscript text (line numbers 492-498). Also, the authors thank the reviewer for endorsing the lower percentage of MCs in the synovial fluid as agreed by the other authors (47, 48, 49).

Reviewer 3 Report

The manuscript: “Mast cells differentiated in synovial fluid and resident in osteophytes exalt the inflammatory pathology of osteoarthritis”.

The title does not reflect this manuscript. Here, the authors focused on the transcriptomics analysis of osteophytes vs controls, in vitro differentiation of ThP1 cells and hematopoietic stem cells treated with OA synovial fluid and proteomics of OA synovial fluid.

In general, the manuscript is difficult to read and follow. There are several points that needs to be explained and justified.

My comments are as follows:

  1. Abstract should be rewritten reporting a brief introduction, aim of the study, methods, results and conclusions.
  2. An introduction on OA pathology involving all joint tissues (cartilage and meniscal degeneration, inflammation and fibrosis of both synovial membrane and infrapatellar fat pad, and subchondral bone remodeling) should be provided considering that the journal is multidisciplinary.    
  3. In general, there are several sentences without references. The authors should check and add appropriate references (for example lines 35-38, 41-43,44-48, 55-56).
  4. Lines 49-54: could the authors better explain this part?
  5. Line 92: why did the authors consider only up-regulated genes?
  6. Figure 1: could the authors define the abbreviations used?
  7. Lines 150-157 and lines 164-167: this part should be moved to the discussion. The authors should report only the results obtained in the results section.
  8. The validation of RNA-seq data is totally unclear. Validation should be performed comparing qRT-PCR (of up-regulated genes found by RNA-seq) betwenn osteophytes and controls.
  9. The rationale of using primary cells isolated from osteophytes to validate the transcriptomics data should be clearly justified. The authors did not validate that these gene are up-regulated. Protocols of primary cell isolation should be reported. What cell type did the authors isolate? How many patients were enrolled for this part? What criteria were of enrollment were used? 
  10. Lines 189-198: could the authors report a graph regarding the quantification performed using IHC comparing control vs patients?
  11. Lines 215-217: Statistical analysis methodology should be moved at the end of the methods. Post hoc test used for multiple comparison is missing.
  12. Lines 219-220: “grade-I to IV” is not clear. Did the authors mean Kellgren-Lawrence? This should be explained. Demographic details of the patients should be added (age, BMI and K-L). How many samples of synovial fluid of grade I, II, III and IV were used to treat ThP1 and HSCs?
  13. Figure 4: this figure should be improved because it is very difficult to understand and the results obtained should be explained in the results section. It seems that the authors use PMA as control. This point should be explained in the results and figure caption as well as protocol should be added in the methods. The authors reported in the caption that they performed histograms from two independent studies. However, they reported in the text of the manuscript that all the flow-cytometry experiments were performed in the triplicates and repeated for three times (lines 214-215). The authors should report and analyze all the data.
  14. The methods should be improved reporting all the necessary information to understand and replicate the experiments.
  15. Why did the authors decide to focus on osteophytes of condyles of tibia? 
  16. Inclusion and exclusion criteria of patients who underwent TKR enrollment should be added. Did these patients sign informed consent? Was this study approved by a ethical committee? Demographic details of the patients should be added (age, BMI and K-L).
  17. Line 430-431: “Control tissue was non-osteophytic bony tissue obtained from lateral condyle of the tibia.” Non-osteophytic bone tissues is too vague. Did the authors collect subchondral bone, cortical or trabecular bone?
  18. Lines 456-474: this part should be deleted.  The authors should report that they followed the manufacturer's protocol.
  19. Did the authors use the 2^-ddct method for the qRT-PCR data quantification?
  20. Lines 525, 554 and 562: H2O2 and CO2 should be corrected. 
  21. Line 557: the authors reported that they changed the medium every three days. Did the authors add 10% synovial fluid every three days?
  22. How did the authors select the synovial fluids used to treat the cells? What criteria did the authors use? Demographic details of these patients should be added (age, BMI and K-L).
  23. How did the authors select the 16 synovial fluids used for proteomic analysis? What criteria did the authors use? Demographic details of these patients should be added (age, BMI and K-L).
  24. The discussion is too long. It should be better focused.

Author Response

Reviewers’ response to Reviewer 3:

Comment 1: Abstract should be rewritten reporting a brief introduction, aim of the study, methods, results and conclusions.

Authors’ response: As per the reviewer’s suggestion, a structured abstract including a brief introduction, aim of the study, methods, results and conclusions is provided now.

Comment 2: An introduction on OA pathology involving all joint tissues (cartilage and meniscal degeneration, inflammation and fibrosis of both synovial membrane and infrapatellar fat pad, and subchondral bone remodeling) should be provided considering that the journal is multidisciplinary.  

Authors’ response: The authors have now added a separate paragraph at the beginning of introduction to provide a general background/view of osteoarthritis and its pathology (line numbers 82-92).

Comment 3: In general, there are several sentences without references. The authors should check and add appropriate references (for example lines 35-38, 41-43,44-48, 55-56).

Authors’ response: The authors thank the reviewer for pointing out this issue. We have now added the appropriate references wherever required.

Comment 4: Lines 49-54: could the authors better explain this part?

Authors’ response: The authors have now elaborately explained the suggested lines (line numbers 112-120).

Comment 5: Line 92: why did the authors consider only up-regulated genes?

Authors’ response: The authors believe that a connection of mast cells and osteophytes is an untouched topic in the research field and explored it further using immunohistochemistry and flow-cytometry studies. Thus, in order to maintain the length and focus of the manuscript on this connection, the authors restricted themselves to consider only up-regulated genes.

Comment 6: Figure 1: could the authors define the abbreviations used?

Authors’ response: All the abbreviations in the Figure 1 are now defined in their full form.

Comment 7: Lines 150-157 and lines 164-167: this part should be moved to the discussion. The authors should report only the results obtained in the results section.

Authors’ response: The authors have implemented the suggested change. Lines 150-157 are moved to discussion section now (line numbers 422-430), while lines 164-167 are removed in the revised version of the manuscript text.

Comment 8: The validation of RNA-seq data is totally unclear. Validation should be performed comparing qRT-PCR (of up-regulated genes found by RNA-seq) betwenn osteophytes and controls.

Authors’ response: Considering the obvious restrictions for the collection of control tissue, availability of control RNA was limited. We therefore could not use control tissue RNA for the validation. This validation was limited to confirm the expression of key upregulated genes from the transcriptomics data in the osteophytes. This fact has been now clarified in the qRT-PCR validation protocol (line numbers 211-212).

Comment 9: The rationale of using primary cells isolated from osteophytes to validate the transcriptomics data should be clearly justified. The authors did not validate that these gene are up-regulated. Protocols of primary cell isolation should be reported. What cell type did the authors isolate? How many patients were enrolled for this part? What criteria were of enrollment were used?

Authors’ response: The authors accept this as mistake and apologize, after clarification and revisiting the records, it is confirmed that the osteophyte tissues were used for qRT-PCR and not the primary cells.

Comment 10: Lines 189-198: could the authors report a graph regarding the quantification performed using IHC comparing control vs patients?

Authors’ response: Anti-FCERI and anti-TPSAB1 staining incidence (%) in control as well as osteophyte samples is now denoted as graphically pie charts in Figure 4 of the revised version of manuscript.

Comment 11: Lines 215-217: Statistical analysis methodology should be moved at the end of the methods. Post hoc test used for multiple comparison is missing.

Authors’ response: The lines 215-217, indicating statistical analysis of in vitro assays has been moved to ‘methods’ section. The authors used Bonferroni’s test for multiple comparison after one-way ANOVA analysis. The related description is now added in methods (line numbers 271-273). 

Comment 12: Lines 219-220: “grade-I to IV” is not clear. Did the authors mean Kellgren-Lawrence? This should be explained. Demographic details of the patients should be added (age, BMI and K-L). How many samples of synovial fluid of grade I, II, III and IV were used to treat ThP1 and HSCs?

Authors’ response: Yes, SF grade-I to IV denotes Kellgren-Lawrence (KL) score of the samples. KL score was used to determine OA severity as well as SF grading (line numbers 155-160 and line numbers 167-168). The details of KL score have been explained in the methodology section (line numbers 155-160). The authors used 3 SF samples of each grade to treat ThP1 and HSCs during the differentiation assay; thus, in total (4 X3) 12 SF samples were used (line numbers 254-255). Demographic details of the patients from whom the SF samples were obtained has been added as ST3 into supplementary data of the manuscript. The patients 1-12 from this table were used for the cell differentiation assay. 

Comment 13: Figure 4: this figure should be improved because it is very difficult to understand and the results obtained should be explained in the results section. It seems that the authors use PMA as control. This point should be explained in the results and figure caption as well as protocol should be added in the methods. The authors reported in the caption that they performed histograms from two independent studies. However, they reported in the text of the manuscript that all the flow-cytometry experiments were performed in the triplicates and repeated for three times (lines 214-215). The authors should report and analyze all the data.

Authors’ response: Please note that Figure 4 is now Figure 5 in the revised version of manuscript. In light of the reviewer’s comment, the methodology and the results of this particular assay has been simplified to understand (line numbers 256-258 and line number 360, respectively). Yes, PMA treated cells were used as a positive control for both the cell types and untreated cells were used as negative control. The relevant description is available in the ‘methodology’, ‘results’ and figure caption too. The flow-cytometry experiments on both the cell types were performed in triplicate and were repeated for three times. As reflected in the figure caption, the histogram presented in the Figure 5 (a, b and c) are representative histogram of each for CD206, HLA-DR and FCERI, respectively. On the other hand, the bar-graphs a1, b1 and c1 presents average results and statistical analysis of all experiments for CD206, HLA-DR and FCERI, respectively.

Comment 14: The methods should be improved reporting all the necessary information to understand and replicate the experiments.

Authors’ response: The methods section has been modified in light of the reviewer’s comment. 

Comment 15: Why did the authors decide to focus on osteophytes of condyles of tibia? 

Authors’ response: As per the review of literature, tibial osteophytes are the feature of osteoarthritis (Hayeri MR et al., 2010), which originates through the pathological changes in subchondral bone of the tibia (Wang Y et al., 2005). Further, they also act as an indicator to determine the disease severity (Kishve and Motwani, 2020). The conclusion from the mentioned and the similar studies helped the authors to focus on the osteophytes from tibial plateau in order to study the molecular events of osteoarthritis associated osteophytes. The reason to focus on the osteophytes particularly from the tibia is now added in the ‘methods’ (line numbers 150-154). 

Comment 16: Inclusion and exclusion criteria of patients who underwent TKR enrollment should be added. Did these patients sign informed consent? Was this study approved by an ethical committee? Demographic details of the patients should be added (age, BMI and K-L).

Authors’ response: Inclusion and exclusion criteria of patient selection for TKR is now described in supplementary data as SD1. The study protocols used for this study were approved by the Institutional Ethics Committee and all the approval numbers are mentioned in the Ethical statement at the beginning of the ‘methodology’ section (142-145).

Comment 17: Line 430-431: “Control tissue was non-osteophytic bony tissue obtained from lateral condyle of the tibia.” Non-osteophytic bone tissues is too vague. Did the authors collect subchondral bone, cortical or trabecular bone?

Authors’ response: Non-osteophytic / control tissue was obtained from epiphyseal trabecular bone from lateral condyle of tibia. This information has been clearly added into osteophytes collection method in ‘methods’ section (line numbers 152-153).

Comment 18: Lines 456-474: this part should be deleted.  The authors should report that they followed the manufacturer's protocol.

Authors’ response: The authors thank the reviewer for his/her suggestion. The suggestion has been implemented now (line number 180).

Comment 19: Did the authors use the 2^-ddct method for the qRT-PCR data quantification?

Authors’ response: Yes, we used comparative or ΔΔCt method of qPCR analysis, the Ct values of different experimental RNA samples are normalized to β-actin, the housekeeping gene. Y-axis denotes relative mRNA abundance.

Comment 20: Lines 525, 554 and 562: H2O2 and CO2 should be corrected. 

Authors’ response: The suggested mistakes are corrected now as - H2O2 and CO2 (line numbers 227, 252, 261).

Comment 21: Line 557: the authors reported that they changed the medium every three days. Did the authors add 10% synovial fluid every three days?

Authors’ response: Yes, 10% synovial fluid was maintained all through the treatment span of 9 days.

Comment 22: How did the authors select the synovial fluids used to treat the cells? What criteria did the authors use? Demographic details of these patients should be added (age, BMI and K-L).

Authors’ response: The synovial fluids used to treat the cells were of all OA grades (KL grade – I, II, III and IV) and were selected randomly. However, For the synovial fluid collection, inclusion and exclusion criteria was applied as described in the supplementary material. Demographic details of the patients from whom these fluids were obtained, is also provided in the supplementary material.

Comment 23: How did the authors select the 16 synovial fluids used for proteomic analysis? What criteria did the authors use? Demographic details of these patients should be added (age, BMI and K-L).

Authors’ response: SF samples of all KL grades (4 SF samples of each KL grade; total 16 SF samples) were randomly selected from the SF repository, collected by the authors. The patient selection criteria for SF collection are described in supplementary data of the manuscript as SD2. Demographic details of the patients from whom SF were obtained and subjected to proteomics analysis has been given in supplementary data as ST4.

Comment 24: The discussion is too long. It should be better focused.

Authors’ response: The discussion section has been revised as guided by the reviewer’s comment.

Round 2

Reviewer 3 Report

The manuscript improved after the revision. However, I have still few comments for the authors.

In the introduction on OA, there is still no mention about the role of the infrapatellar fat pad.  

The authors should explain in the text of the manuscript that why they focused only on up-regulated genes.

The authors performed RNA-seq on tibial osteophytes vs epiphyseal trabecular bone from condyle of tibia of six patients. Regarding the validation, which is fundamental, the authors replied to me that it was not possible to have control tissue RNA. Few points are not clear to me: 1) how many patients were enrolled for RNA-seq validation? Are these patients the same 6 patients used for RNA-seq?  2) Why did the author not isolate RNA from the epiphyseal trabecular bone of the tibial condyle as control?

Supplier of all reagents should be specified. 

At the end of the methods, a paragraph regarding statistical analysis is mandatory.

If the authors used the same 6 patients or if it is not possible to compare gene expression in osteophytes vs control, this should be added in the limitation of the study.  

Figure 3: Statistical analysis is unclear. Are MMP-1, MMP-3 etc statistical different compared to ACTB? This should be reported as follow: “ * P < 0.05, ** P < 0.01 compared to ACTB.”.  

Figure 4: Pie chart are not the best way to represent these kind of data. Normally histograms are used.  A histogram comparing Anti-FCERI in control vs patients and a histogram comparing anti-TPSAB1 in controls vs patients should be added, statistical analysis should be performed and reported.

Figure 5: Statistical analysis is unclear. G-1, G-II etc. are statistically different compared to UC or PMA in fig. a1?

Author Response

A response to the reviewer queries has been submitted as a separate word document. 
